# Transcranial Direct Current Stimulation (tDCS): A Promising Treatment for Major Depressive Disorder?

**DOI:** 10.3390/brainsci8050081

**Published:** 2018-05-06

**Authors:** Djamila Bennabi, Emmanuel Haffen

**Affiliations:** Department of Clinical Psychiatry, CIC-1431 INSERM, CHU de Besançon, EA Neurosciences, University Bourgogne Franche-Comte, FondaMental Foundation, 94000 Creteil, France; emmanuel.haffen@univ-fcomte.fr

**Keywords:** transcranial direct current stimulation, depression, cognition

## Abstract

**Background:** Transcranial direct current stimulation (tDCS) opens new perspectives in the treatment of major depressive disorder (MDD), because of its ability to modulate cortical excitability and induce long-lasting effects. The aim of this review is to summarize the current status of knowledge regarding tDCS application in MDD. **Methods:** In this review, we searched for articles published in PubMed/MEDLINE from the earliest available date to February 2018 that explored clinical and cognitive effects of tDCS in MDD. **Results:** Despite differences in design and stimulation parameters, the examined studies indicated beneficial effects of tDCS for MDD. These preliminary results, the non-invasiveness of tDCS, and its good tolerability support the need for further research on this technique. **Conclusions**: tDCS constitutes a promising therapeutic alternative for patients with MDD, but its place in the therapeutic armamentarium remains to be determined.

## 1. Introduction

Major depressive disorder (MDD) is a widespread psychiatric disease characterized by a high risk of morbidity and mortality and a high level of comorbidity with several psychiatric and non-psychiatric disorders [1]. Epidemiological surveys have repeatedly indicated a high lifetime prevalence of this illness, amounting to 6.7% of the population worldwide [2]. The costs of MDD to society, in terms of personal and familial suffering and health care consumption, are high [3,4]. Despite the progress in pharmacopoeia and in psychological therapies, clinicians involved in the management of MDD are regularly faced with clinical situations marked by treatment resistance. About 30% of depressed patients fail to experience significant clinical benefits from currently available treatments [5,6], leading to a chronically deteriorating course of the illness. Consequences of the illness and limitations of the usual pharmacological and psychological strategies highlight the necessity to develop alternative therapeutic options.

Brain stimulation therapies have emerged as relevant alternative strategies, on the basis of emerging knowledge about specific brain areas involved in psychiatric diseases. Among these techniques, transcranial direct current stimulation (tDCS) appears to hold particular promise because of its cost, ease of use, and favorable tolerability profile [7]. tDCS was used from the 1960s to generate modifications of cortical excitability in preclinical studies and as a therapeutic tool for MDD. More recently, this technique has gained renewed interest as a practical tool for the modulation of cortical excitability and the treatment of psychiatric disorders [7,8]. tDCS relies on the application of a weak direct current of 1–2 mA directly to the scalp through electrodes to induce regional changes in cortical excitability that can last up to a few hours after stimulation [9]. Sustainable effects seem to be mediated by bidirectional modifications of postsynaptic connections similar to long-term potentiation and depression, occurring through NMDA-dependent mechanisms [10,11]. Because of the implication of pathological alterations of neuroplasticity in psychiatric disorders, tDCS appears to be a promising therapeutic alternative to modify such pathological plasticity [12]. Beneficial effects of tDCS have been reported in the treatment of psychiatric (mainly depression and schizophrenia) and neurological diseases, as well as in the rehabilitation of cognitive, motor, and sensory functions after a stroke [8,13].

In this review, we summarize data obtained from trials with the aim of assessing the effectiveness of tDCS as a putative treatment option for MDD. 

### 1.1. Method

To identify studies reporting the effects of tDCS on depression and cognition in MDD, two authors (E.H. and D.B.) searched the Pub-med database following PRISMA recommendations [14]. The identification of articles was based on the following keywords: “depression”, “transcranial direct current stimulation”. 

Inclusion criteria for this review were: (a) publication in English; (b) inclusion of depressed patients treated with tDCS; (c) meta-analysis, randomized controlled studies (RCTs) and open-label trials; (d) assessment of depressive symptoms and/or cognition. Working independently and in duplicate, the two reviewers examined all titles and abstracts, obtained full texts of potentially relevant papers, and read the papers to determine whether they met the inclusion criteria. Of the 381 initial references, 67 papers were retained. We excluded seven studies exploring the effects of tDCS in healthy subjects, one review, and one protocol. Thus, we obtained data from 58 articles that met our eligibility criteria.

### 1.2. Technical and Safety Aspects of Transcranial Direct Current Stimulation

During tDCS sessions, a constant direct current of low intensity (generally 1–2 mA) is passed across the brain via electrodes wrapped in an electrode gel or saline-soaked sponge pockets. [8]. Wide variations in the amount of current delivered can be observed according to the stimulation parameters applied (i.e., position and size of the electrodes, current intensity, duration, frequency, and number of sessions). In the majority of protocols, the currents ranged from 0.5 to 2 mA and were delivered for 5–30 min via electrodes of 25–35 cm, generating current densities of 0.28–0.8 mA/cm^2^. The placement of the electrodes is typically based on the 10–20 electrode placement system to locate the area of interest. More recent studies suggest the use of an initial magnetic resonance imaging (MRI) scan to refine the position of the electrode, taking into account the inter-individual variability [15]. From a neurobiological standpoint, tDCS is considered a technique of neuromodulation, given that it does not directly induce action potentials, unlike transcranial magnetic stimulation. The polarity of the stimulation determines the type of effect; anodal stimulation induces a depolarization of the neuronal membranes and an increase of the spontaneous neuronal firing rate, whereas cathodal stimulation leads to neuronal hyperpolarization and inhibition [16,17]. Depending on the duration and intensity of the stimulation, it has been shown that the modulation of electrophysiological properties is directly measurable in healthy volunteers and could last up to 90 min after stopping the stimulation [9]. 

tDCS may induce mild to moderate side effects, including light itching beneath the electrodes, mild headache, tingling, burning sensations, and discomfort. Skin irritation and lesions under the electrodes have also been reported, as well as some cases of mood switching [18,19].

### 1.3. Effects of tDCS on Symptoms

The rationale for using tDCS in this indication is based on historical observations of hypometabolism of the left dorsolateral prefrontal cortex (dlPFC) associated with right prefrontal hyperfunction in MDD and dysfunction of brain plasticity, characterized by an alteration of long-term potentiation and depression [20,21]. In the majority of the protocols, the currents used were 1 or 2 mA and were applied for 30 min via two large conductive electrodes (32–35 cm^2^) soaked with a saline solution. The anode was typically placed over the left dlPFC, and the cathode over the contralateral supraorbital area, corresponding to F3 and FP2 according to the international 10–20 EEG system [22]. Transcranial direct current stimulation was proposed either alone or as an add-on treatment to psychotropic medication or cognitive training programs. 

Following the seminal work of Fregni et al. [23] which produced positive results on the efficacy of five sessions of anodal tDCS applied over the left dlPFC (1 mA, 10 min) in ten patients with MDD, a series of controlled and open-label studies have been published. Subsequently, two RCTs conducted by Boggio et al. [24,25] have replicated these results. The authors observed an average reduction of 40.4% in the Hamilton Depression Rating Scale (HDRS) score after anodal tDCS (2 mA, 20 min) versus 10.4% after placebo stimulation in 40 patients with mild to moderate MDD. In 2010, Loo et al. [26] showed no superior antidepressant efficacy of ten sessions of tDCS (1 mA, 20 min) versus placebo in 40 patients with MDD. However, in a second study, these authors demonstrated a greater decrease in the Hamilton Depression Rating Scale (HDRS scores in 64 unipolar and bipolar depressed patients after 15 sessions of anodal tDCS of the left dlPFC (2 mA, 20 min) (28.4%) versus placebo (15.9%) [27]. Rigonatti et al. [28] compared the effect of fluoxetine and ten tDCS sessions (2 mA, 20 min) in 42 depressed patients, and noted a similar improvement in depressive symptoms following brain stimulation and pharmacological treatment, with an earlier antidepressant action in the tDCS group. In the field of treatment-resistant depression, three RCTs were conducted and did not show any significant difference between active stimulation and placebo [29,30,31]. Regarding post-stroke depression, Valiengo et al. [32] studied the effects after active versus sham stimulation in 45 patients and reported a higher response rate with active stimulation (37.5% and 20.8%, respectively) and higher remission rates (4.1% and 0%, respectively). 

In this context, in 2013, a larger study was conducted involving 120 unipolar depressed patients, which compared tDCS versus a pharmacological treatment (sertraline) and versus tDCS plus sertraline. The results showed a greater reduction in Montgomery–Asberg Depression Rating Scale scores in patients receiving the combined intervention (tDCS + sertraline) versus those receiving sertraline alone (mean difference 8.5 points), tDCS alone (mean difference 5.9 points), or placebo (mean difference 11.5 points) [33]. More recently, these authors published the results of a non-inferiority trial involving 245 depressed patients and compared active tDCS (2 mA, 30 min) versus a pharmacological treatment (escitalopram) and versus placebo. They concluded that escitalopram was significantly superior to tDCS [34]. Moreover, a large-scale RCT involving 84 unipolar and 36 bipolar depressed patients reported comparable effects of active (2.5 mA, 30 min, 20 sessions over four weeks) and sham tDCS applied over the left dlPFC, suggesting that the low level of stimulation delivered in sham conditions may have been biologically active [35].

Studies evaluating the effectiveness of tDCS in a maintenance therapy for relapse prevention have shown that the reduction in treatment frequency from two to one week or a high level of pre-treatment resistance was associated with a greater rate of relapse [36,37].

Several meta-analyses pooling available data about the antidepressant efficacy of tDCS have yielded contradictory results. Kalu et al. [38] showed a higher efficacy of active tDCS versus placebo, as evidenced by a greater reduction in severity scores on depression scales. Berlim et al. [39] did not find any significant difference between active tDCS and placebo in response rates (23.3% versus 12.4%, respectively, *p* = 0.11) and remission (12.2% versus 5.4%, respectively, *p* = 0.22). Shiozawa et al. [40] included seven controlled trials in their meta-analysis and showed a superiority of active tDCS over placebo in terms of clinical response and remission, which was confirmed by Meron et al. [41] in a meta-analysis of 10 RCTs. In a meta-analysis of individual data from 289 patients, Brunoni et al. [42] demonstrated the superiority of active tDCS compared to placebo in terms of alleviation of depressive symptoms, with a response rate of 33.3% versus 19 %, respectively, and a remission rate of 23.1 versus 12.7%, respectively. The level of response was correlated with various parameters, namely, the number of sessions and the amount of energy delivered, but was inversely associated with the level of antidepressant resistance. Other variables, such as the severity of the current depressive episode, the presence of bipolar disorder, female gender, or treatment with sertraline, as well as pre-treatment motor retardation or better verbal fluency, were also identified as potential predictors of a positive response [42,43,44,45]. From a neurobiological point of view, 5-HTTLPR polymorphism has shown a predictive property, while brain-derived neurotrophic factor, cytokine levels, neurotrophins 3 and 4, nerve growth factor, and glial cell line-derived neurotropic factor have failed to predict the clinical response [46,47,48,49]. 

### 1.4. Effects of tDCS on Cognition

Current evidence suggests that tDCS may have an impact on some cognitive functions in healthy volunteers, such as working memory, attentional performance, procedural learning, and emotional information processing [50]. In MDD, although most studies reported an improvement in at least some cognitive functions, suggesting a potential pro-cognitive role of tDCS, no firm conclusions could be drawn. To date, the improvement of attention and working memory has been reported after 1 [27,51,52], 5 [53], and 10 [25] anodal tDCS sessions applied over the left DLPFC. Positive results have also been observed in other cognitive domains, such as cognitive control [54,55], processing speed [56], or emotion recognition [57]. Bifrontal tDCS has been shown to promote more accurate and faster responses to the n-back task, exploring working memory, and to prevent procedural learning during the probabilistic classification learning task in depressive states [51]. Moreover, a few clinical cases of improvement in cognitive performances after treatment with tDCS have been reported in the context of treatment-resistant depression or post-traumatic depression [29,58]. However, several RCTs applying a set of standardized cognitive tests did not record tDCS-related changes with offline stimulations, suggesting that repeated sessions have no cumulative effects [26,31,59]. Beyond the variability in the stimulations parameters (i.e., site of stimulation or use of off- or online sessions) and the impact of inter- and intraindividual differences, in most of the studies it is difficult to differentiate the “pure” effects of cognitive improvement of tDCS from a “collateral” effect relied on a cognitive improvement due to the alleviation of depression [60].

### 1.5. Parameters Influencing Clinical Outcomes

Multiple factors are likely to modulate the therapeutic effects of tDCS. Among them, the stimulation parameters and clinical characteristics of the population are key contributors to the variability of its effects [60,61,62]. With regard to MDD, there is a dearth of clinical trials exploring the impact of the stimulation parameters on clinical outcomes. Typically, bifrontal montages (F3–F8 and F3–F4 montages) targeting the left dlPFC are used in MDD. However, a computational modelling study suggested that other montages, using a fronto-extracephalic or fronto-occipital approach, could result in greater stimulation of brain regions of critical interest, such as the anterior cingulate cortex, which may be advantageous for treating MDD [63]. In fact, two open-label trials observed improvement in depressive symptoms following tDCS sessions applied over the fronto-occipital or -temporal regions [64,65] in a total of 18 patients and 20 sessions. Moreover, combined with sertraline, tDCS applied for 20 or 30 min was found to be effective for the treatment of mild and moderate depression, and the effect of the stimulation for 30 min was more pronounced than that of the 20 min stimulation [66]. Meta-analyses have noted that increasing the number of sessions or the intensity of the stimulation (1 versus 2 mA) does not improve tDCS effects on depressive symptoms [38,39]. In addition, the delay between sessions could have an impact on tDCS effects [67]. For example, Alonzo et al. [68] reported that daily tDCS induced a greater increase in cortical excitability than second daily stimulation of the motor cortex. Another, often overlooked, point is the influence of patient characteristics. Two open-label trials reported that depression severity was positively related to clinical improvement [69,70]. Evidence from three RCTs indicated that dlPFC tDCS had lesser efficacy in treatment-resistant depressed patients [29,31]. Finally, a meta-analysis of data from seven RCTs showed the superiority of active tDCS versus placebo in bipolar depression in terms of reduction in severity scores on depression scales from the first week of treatment [71]. Besides the features of the ongoing episode, the effects of interindividual variables, such as anatomical differences, genetic factors, personality, comorbidities, cognitive strategy, lifestyle, and baseline neuronal activation state, need to be explored [50]. Likewise, the age at which the stimulation is delivered might be a key determinant of the physiological and behavioral outcomes of tDCS. In children and adolescents, specific effects of tDCS on cortical excitability have been demonstrated [72,73]. In this population, the choice of the stimulation parameters, especially the dose selection, requires special attention [74]. 

It should also be emphasized that the final effects of tDCS depend on the concomitant use of pharmacotherapy with other interventions, such as cognitive therapy. A synergistic therapeutic action was observed with the combination of tDCS and sertraline in MDD patients in a large-scale RCT which compared the efficacy of tDCS, sertraline, and a combination of both [33]. Conversely, benzodiazepines were reported to reduce tDCS effects [75]. Concerning the effects of adjunctive tDCS and cognitive control therapy, Segrave et al. [76] showed that active tDCS coupled with weekly cognitive behavioral therapy (CBT) potentiated the treatment response, while Brunoni et al. [77] failed to demonstrate the superiority of combined cognitive control training (CCT) and active tDCS intervention versus CCT and sham tDCS. In resistant depression, one open study reported an improvement in depressive symptoms after 18 sessions of tDCS over six weeks, administered during Cognitive Emotional Training [78]. More recently, Mayur et al. failed to demonstrate differences in terms of speed of response or cognitive performance after the use of tDCS in combination with electroconvulsive therapy (ECT) versus ECT alone [79].

## 2. Conclusions

tDCS is a promising therapeutic strategy that offers the opportunity for non-invasive modulation of cortical excitability and plasticity in psychiatric disorders. With regard to MDD, the majority of meta-analyses have found that tDCS is superior to sham stimulation with an effect size (B coefficient = 0.35) comparable to that of repetitive transcranial magnetic stimulation (rTMS) and antidepressant medication in primary care [42]. This technique appears to be particularly indicated for patients with a mild-to-severe form of MDD who do not meet the criteria for resistant depression. In line with these data, a European expert group has recently proposed a Level B recommendation for its use in depressed patients, treated or not with antidepressants, without treatment resistance (minimum of 10 sessions (2 mA, 20–30 min) with the anode over the left dlPFC and the cathode over the right supra-orbital region) [7]. Questions still remain unanswered regarding the optimal stimulation parameters, the effect of tasks given during tDCS sessions, and the possible influence of add-on medications. Moreover, the clinical profile of depressed patients showing favorable responses to tDCS require clarification. Besides these critical questions, the promising preliminary results, the non-invasiveness of tDCS, and its good tolerability support the need for further research into this technique.

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
