# Peer review of "Transcranial Direct Current Stimulation (tDCS): A Promising Treatment for Major Depressive Disorder?"

_brainsci, 2018, doi:10.3390/brainsci8050081_

Round 1

Reviewer 1 Report

My main problem with this review is that it lacks novel insight or ideas in the field of treating major depression with tDCS. There are a variety of articles and even reviews citing the same trials and coming to similar conclusions (see below), so I doubt the present manuscript could make a novel contribution at the moment - and, to be honest, I do not see the point of generating several review articles at the same time (maybe apart from collecting citations since review articles are cited a lot?). Therefore I do not recommend it for publication, but of course, the final decision has to me made by the editor.

List of similar review articles (with similar data and conclusions and open questions) that have been published recently:

https://www.ncbi.nlm.nih.gov/pubmed/29066167 (2018)

https://www.ncbi.nlm.nih.gov/pubmed/29163106 (2017)

https://www.ncbi.nlm.nih.gov/pubmed/29294333 (2017)

https://www.ncbi.nlm.nih.gov/pubmed/27916405 (2017)

https://www.ncbi.nlm.nih.gov/pubmed/27866120 (2017)

https://www.ncbi.nlm.nih.gov/pubmed/27885309 (2016)

https://www.ncbi.nlm.nih.gov/pubmed/27056623 (2016)

https://www.ncbi.nlm.nih.gov/pubmed/26232699

https://www.ncbi.nlm.nih.gov/pubmed/24713139

Author Response

The authors are grateful for the reviewer's comments.

We are agree with the fact that there are several review articles in this field. However, the reviews articles cited are covering different topics (effects of medication, case of mania or hypomania...). Moreover, the field is quickly  evolving, the data gathered and contained within this manuscript builds on findings from previous reports and provides additional and most recent references.

Reviewer 2 Report

This well-written review provides a succinct, though comprehensive overview of tDCS treatment for major depressive disorder which is accessible to the general reader. I have only the following comments about the paper in its current form:

Section 1.2. Please provide references for the statement beginning line 76: "The polarity,,,"

Page 3, line 126. The reference Valiengo et al. (2013), Dep and Anxiety, is missing from this sentence.

Page 3, line 141. Martin et al. (2016) found that better verbal fluency predicted tDCS response.

Page 4, line 152. For reference 25, cognitive enhancing effects were found after 1 tDCS session, not 15.

Page 4, line 157. Reference 49 is incorrect for this statement.

Page 5, line 201. Segrave and Brunoni investigated the combination of tDCS with cognitive control training, not CBT.

Page 5, line 219. Could add here that another answered issue is about concurrent tasks given during tDCS for MDD.   

Author Response

The authors are grateful for the reviewer's comments

Section 1.2. Please provide references for the statement beginning line 76: "The polarity,,,"

We added the following references:

Nitsche, M.A.; Cohen, L.G.; Wassermann, E.M.; Priori, A.; Lang, N.; Antal, A.; Paulus, W.; Hummel, F.; Boggio, P.S.; Fregni, F.; Pascual-Leone, A. Transcranial direct current stimulation: state of the art 2008. Brain Stimul 2008, 1, 206–223, doi:10.1016/j.brs.2008.06.004.

Nitsche, M.A.; Doemkes, S.; Karaköse, T.; Antal, A.; Liebetanz, D.; Lang, N.; Tergau, F.; Paulus, W. Shaping the effects of transcranial direct current stimulation of the human motor cortex. J. Neurophysiol. 2007, 97, 3109–3117, doi:10.1152/jn.01312.2006.

Page 3, line 126. The reference Valiengo et al. (2013), Dep and Anxiety, is missing from this sentence.

We added this reference.

Page 3, line 141. Martin et al. (2016) found that better verbal fluency predicted tDCS response.

We modified the sentence: "Other variable, such as the severity of the current depressive episode, the presence of bipolar disorder, female gender or treatment with sertraline, as well as pre-treatment motor retardation or better verbal fluency, were also identified as potential predictors of a positive response ".

Page 4, line 152. For reference 25, cognitive enhancing effects were found after 1 tDCS session, not 15.

We modified the sentence:"To date, the improvement of attention and working memory has been reported after one [27,51,52], five [53] and ten [25] anodal tDCS sessions applied over the left DLPFC".

Page 4, line 157. Reference 49 is incorrect for this statement.

We modified this reference.

Page 5, line 201. Segrave and Brunoni investigated the combination of tDCS with cognitive control training, not CBT.

We modified the sentence: "Concerning the effects of adjunctive tDCS and cognitive control therapy, Segrave et al. [76] showed that active tDCS coupled with weekly cognitive behavioral therapy (CBT) potentiated treatment response, while Brunoni et al. [77] failed to demonstrate the superiority of combined cognitive control training (CCT)/active tDCS intervention versus CCT/sham tDCS."

Page 5, line 219. Could add here that another answered issue is about concurrent tasks given during tDCS for MDD. 

We added the sentence:"Questions still remain unanswered regarding the optimal stimulation parameters, the effect of tasks given during tDCS sessions and the possible influence of add-on medications".